



# Late Neogene evolution of modern deep-dwelling plankton

Flavia Boscolo-Galazzo[1‡*], Amy Jones[2], Tom Dunkley Jones[2], Katherine A. Crichton[1↓], Bridget S. Wade[3], Paul N. Pearson[1]

[1]Cardiff University, School of Earth and Environmental Sciences, Cardiff (UK).

[2]Birmingham University, School of Geography, Earth and Environmental Sciences, Birmingham (UK).

[3]University College London, Department of Earth Sciences, London (UK).

‡Now at Bergen University, Department of Earth Science and Bjerknes Center for Climate Research, Bergen (Norway).

↓Now at Exeter University, Department of Geography, Exeter (UK)

*corresponding author: flavia.boscologalazzo@uib.no

The fossil record of marine microplankton provides insights into the evolutionary drivers which led to the origin of modern deep-water plankton, one of the largest component of ocean biomass. We use global abundance and biogeographic data combined with depth habitat reconstructions to determine the environmental mechanisms behind speciation in two groups of pelagic microfossils over the past 15 million years. We compare our microfossil datasets with water column profiles simulated in an Earth System model. We show that deep-living planktonic foraminiferal (zooplankton) and calcareous nannofossil (mixotroph phytoplankton) species were virtually absent globally during the peak of the middle Miocene warmth. Evolution of deep-dwelling planktonic foraminifera started from subpolar-midlatitude species during late Miocene cooling, via allopatry. Deep-dwelling species subsequently spread towards lower latitudes and further diversified via depth sympatry, establishing modern communities stratified hundreds of meters down the water column. Similarly, sub-euphotic zone specialist calcareous nannofossils become a major component of tropical and sub-tropical assemblages during the latest Miocene to early Pliocene. Our model simulations suggest that increased organic matter and oxygen availability for





planktonic foraminifera, and increased nutrients and light penetration for nannoplankton, favored the evolution of new deep water niches. These conditions resulted from global cooling and the associated increase in the efficiency of the biological pump over the last 15 million years.

## 1. Introduction

The biodiversity of planktonic and nektonic organisms is difficult to explain given the uniform character and vastness of pelagic environments, where genetic isolation seems difficult to maintain (Norris, 2000). Planktonic microorganisms with mineralized shells have often been used as a model to study the mode and tempo of species origination in the open ocean, due to the abundance, widespread distribution, and temporal continuity of their fossil record (e.g., Pearson et al., 1997; Norris, 2000; Bown et al., 2004; Ezard et al., 2011; Norris et al., 2013). Because of the great fossilization potential of their calcium carbonate tests across much of the global ocean, their relatively simple and well-established taxonomy, and highly resolved biostratigraphy, planktonic foraminifera and calcareous nannofossils are amongst the best studied fossil groups. Planktonic foraminifera are heterotrophic zooplankton, with different species specialized to feed on different types of food, from other plankton to sinking detritus. In the modern ocean, planktonic foraminifera live stratified across a range of depths spanning from the surface to hundreds of meters down the water column (Rebotim et al., 2017; Meilland et al., 2019). Properties such as food quantity and quality, oxygen, light and pressure all change markedly across the first few hundreds of meters of the ocean. Depending on such down-column variability in environmental conditions, planktonic foraminifera can actively control their living depth of preference, which remains relatively stable during their adult life-stage (Hull et al., 2011; Weiner et al., 2012; Rebotim et al., 2017; Meilland et al., 2019; Duan et al., 2021). A key advantage of using planktonic



foraminifera for evolutionary studies is the ability to extract ecological information from their shell
chemistry. This provides invaluable information about species-specific functional ecology (e.g.,
feeding strategy) and habitat preferences (e.g., surface versus deep waters), which in combination
with biogeographic, taxonomic, biometric, and stratigraphic data have often been used to infer
speciation and extinction mechanisms (Norris et al., 1993; Norris et al., 1994; Pearson et al., 1997;
Hull and Norris, 2009; Pearson and Coxall, 2012; Woodhouse et al., 2021 ) and reconstruct
phylogeny (Aze et al., 2011).
Calcareous nannoplankton also have a highly resolved and continuous fossil record; they are
the most abundant microfossils in oceanic pelagic sediments, and similar to planktonic
foraminifera, their spatial distribution ranges from tropical to subpolar latitudes (Poulton et al.
2017). In the modern ocean they also live stratified in the water column, with species adapted to
euphotic waters, and species adapted to live deeper (Poulton et al., 2017). In contrast to planktonic
foraminifera, nannoplankton are predominantly autotrophic, performing photosynthesis in water
where light penetration is sufficient, although there is evidence for heterotrophy (mixotrophic
behavior) in some extant (Godrjian et al., 2020) and fossil (Gibbs et al., 2020) species. In euphotic
waters, organic matter production from nannoplankton is at the base of pelagic food chains and of
the functioning of the ocean biological carbon pump. Taxonomic, biometric and stratigraphic data
have been used to establish phylogenetic relationships between fossil nannoplankton species
(Young and Bown, 1997).
Little emphasis has been given to the long-term drivers of evolutionary patterns observed in
fossil plankton from species to phylum level, although more recently, a broad connection with
changing climate and ocean properties has been suggested (e.g., Ezard et al., 2011; Norris et al.,
2013; Frass et al., 2015; Henderiks et al., 2020; Lowery et al., 2020). Boscolo-Galazzo, Crichton



et al. (2021) showed that over the last 15 million years the remineralization of particulate organic
carbon (POC) in surface waters declined markedly driven by climate and ocean cooling (Kennett
and Von der Borch, 1985; Kennett and Exon, 2004; Cramer et al., 2011; Zhang et al., 2014; Herbert
et al., 2016; Sosdian et al., 2018; Super et al., 2020), increasing the efficiency of the ocean
biological pump in delivering organic matter at depth. Such a mechanism was key to promote the
evolution of life in deep waters, allowing the development of the modern twilight zone ecosystem
(Boscolo-Galazzo, Crichton et al., 2021). The goal of this study is to combine the fossil record of
two ecologically complementary calcareous microplankton groups seldom analyzed together,
planktonic foraminifera and nannoplankton, and together with model simulations, help disentangle
the evolutionary drivers of modern deep-dwelling plankton. We use the planktonic foraminiferal
dataset from Boscolo-Galazzo, Crichton et al. (2021) and extend our analysis to calcareous
nannofossils in coeval sediment samples to assess their abundance and distribution pattern. We
compare the results from these two groups and contrast them against time and site-specific model
water column profiles for POC and oxygen ($O_2$) obtained from the cGENIE Earth System model.
Further, using stable isotopes, depth habitat reconstructions, abundance and biogeography data we
reconstruct the speciation mechanisms which led to the evolution of modern deep-dwelling
planktonic foraminiferal species.

**2. Methods**
**2.1 Planktonic foraminifera**

In this study we focus on the deep-dwelling groups of macroperforate planktonic foraminifera

of the hirsutellids, globorotaliids, truncorotaliids and globoconellids, which in the modern ocean
calcify and live mostly in the twilight zone of the ocean, i.e. between 200-1000 m (Birch et al.,





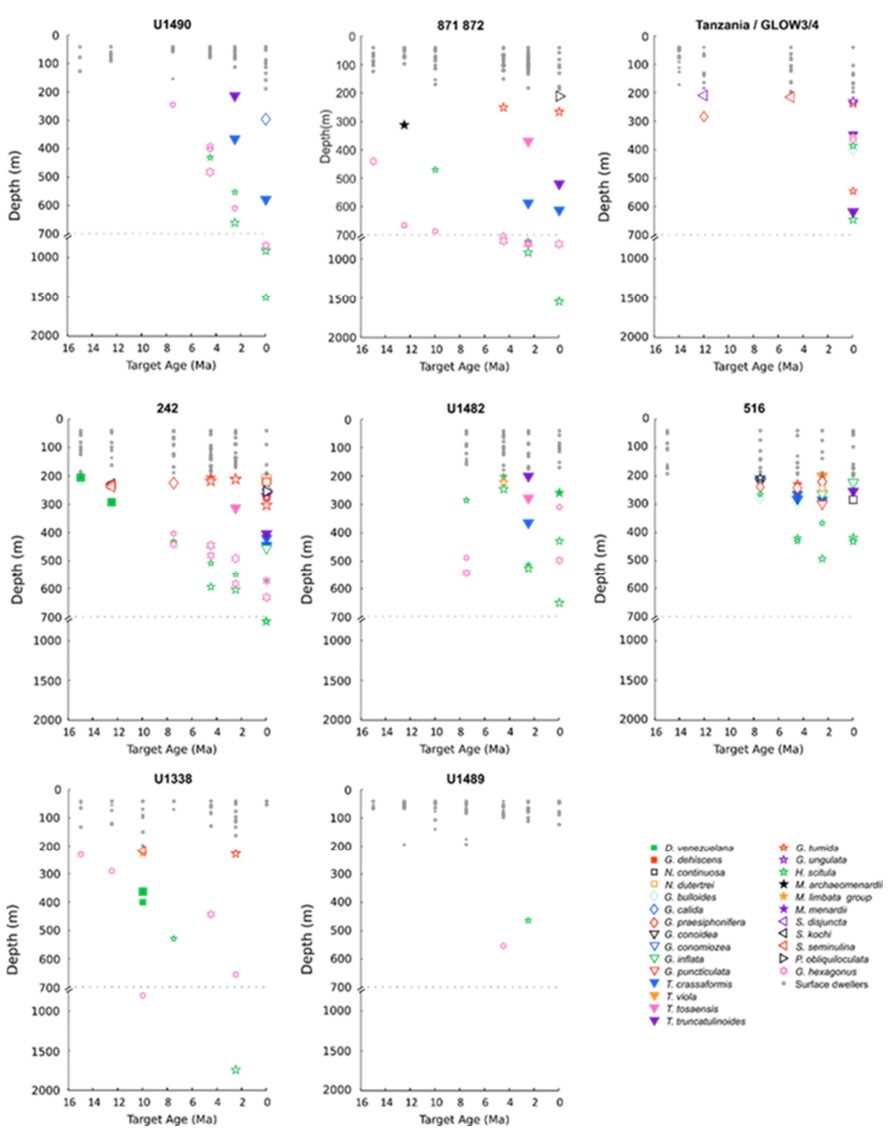

**Figure 1.** Depth-habitat reconstructions for middle Miocene to present planktonic foraminiferal species at the investigated sites. Surface dwellers (species living at depths shallower than 200 m) are indicated with a grey dot, deeper species are indicated with colored symbols. Relative size of symbols represents the size fractions of the sample. Reproduced from Boscolo-Galazzo, Crichton et al. (2021).





2012; Rebotim et al., 2017), and have a more complete fossil record than deep-dwelling
microperforate planktonic foraminifera (Kennett and Srinivasan, 1983).
Planktonic foraminiferal data and depth habitat reconstructions (Fig. 1) are from Boscolo-
Galazzo, Crichton et al. (2021). They were obtained from globally and latitudinally distributed
DSDP/ODP/IODP sites and from cores drilled onshore and offshore Tanzania, all characterized
by abundant calcareous microfossils (Boscolo-Galazzo, Crichton et al., 2021). The work was
focused on seven target ages (15 Ma, 12.5 Ma, 10 Ma, 7.5 Ma, 4.5 Ma, 2.5 Ma, 0 Ma/Holocene).
To avoid sample-aliasing, bulk sediment stable isotopes were measured on an average of ten
samples per target age at each site. The sample displaying mean oxygen stable isotope values was
chosen for subsequent analyses (Boscolo-Galazzo, Crichton et al., 2021). Ages were determined
based on biostratigraphic analysis mostly following the biozonation scheme by Wade et al. (2011).
Foraminiferal picking for stable isotope measurements were conducted from the size fractions:
180-250 μm, 250-300 μm, 300-355 μm (Boscolo-Galazzo, Crichton et al., 2021). Stable isotopes
were measured on an average of 15 different species per sample, using ~25 specimens for common
species, and as many specimens as possible for rare species. Stable isotopes were measured at
Cardiff University. Stable isotope results are shown in Figs. S1 to S9 in the Supplement. Only data
from the largest of the three measured size fractions are shown when data from more than one size
fraction are available. Data from size fractions other than those above are shown only when a
species did not occur within the preferred size interval. Foraminiferal abundance counts were
carried out in two size fractions, 180-250 μm and >250 μm, counting up to 300 specimens in each.
Total abundances were derived by summing up abundances from these two size fractions.
Boscolo-Galazzo, Crichton et al. (2021) reconstructed planktonic foraminiferal depth habitat
(Fig. 1) using a combined model-data approach, solving the paleotemperature equation of Kim and





O'Neill (1997) for each data point using measured foraminiferal $\delta^{18}$O, global ice volume estimates,
and the cGENIE modeled salinity field to determine local water $\delta^{18}$O, and then use the model
temperature-depth curve to determine depth. The full method is described in Boscolo-Galazzo,
Crichton et al. (2021).

**2.2 Calcareous Nannofossils**
Quantitative calcareous nannofossil data were collected from the same samples as used for
planktonic foraminiferal analysis or, when this was not possible, stratigraphically adjacent samples
(Table S1 in the Supplement). A cascading count technique was used to maximise nannofossil
diversity recovery and quantification of low abundance species (Styzen, 1997). Nannofossils were
counted per field of view (FOV) until a minimum of 400 specimens were achieved for each sample.
However, if a high abundance species exceeded an average of 25 specimens per FOV, it was
excluded from subsequent counts in that sample and its abundance scaled-up based on its average
abundance and the total numbers of FOV counted. Only specimens directly counted contributed
to the minimum count threshold of 400 specimens. An additional scan of two slide transects were
undertaken to record rare species not observed during the extended count and are included in the
total species richness and diversity analyses. Samples for nannofossil analysis were prepared using
the smear slide technique (Bown & Young, 1998). Calcareous nannofossils were observed using
both plane-polarised (PPL) and cross-polarised light (XPL) on a Zeiss Axioscope light-microscope
at x1000 magnification. Identification and taxonomy used herein follows Young et al. (1997) and
is coherent with the recent Neogene calcareous nannofossil taxonomy (Ciummelli et al., 2016;
Bergen et al., 2017; Blair et al., 2017; Boesiger et al., 2017; Browning et al., 2017; de Kaenel et
al., 2017).





**2.3 Plankton Ecogroups**
In order to compare the datasets obtained from the planktonic foraminiferal and nannofossil
analysis, we grouped species into ecogroups based on depth-habitat preferences. Planktonic
foraminiferal ecogroups are defined based on paleodepth habitat reconstructions from Boscolo-
Galazzo, Crichton et al. (2021): the euphotic zone ecogroups includes species with an average
depth habitat shallower than 200 m (the bottom of the euphotic zone), the twilight zone ecogroup
includes species with an average depth habitat coinciding with the twilight zone (200-1000 m).
The twilight zone ecogroup is largely composed of species within the globoconellids, the
*Globorotalia merotumida-tumida* lineage, the hirsutellids and the truncorotaliids, but also includes
species from other genera, such as *Globigerinella calida*, *Globorotaloides hexagonus*, *G.
variabilis*, and *Pulleniatina obliquiloculata*. *Dentoglobigerina venezuelana* has a changeable
depth habitat through time (see discussion in Wade et al., 2018); following the depth habitat
reconstructions from Boscolo-Galazzo, Crichton et al., (2021) it was grouped as euphotic zone
species for target ages 15, 12.5, 7.5, 4.5 Ma and as twilight zone species for target age 10 Ma.
Species were excluded from the grouping when they are known to have a marked seasonality in
abundance and depth habitat (*Globigerina bulloides, Globigerinella praesiphonifera* and the
neogloboquadrinids) (e.g., Jonkers and Kucera, 2015; Greco et al., 2019), and if they were too rare
and depth habitat reconstruction was not possible (*Candeina nitida*).
Three ecogroups for calcareous nannofossils are used: upper-euphotic, lower-euphotic and
sub-euphotic. The upper-euphotic group is represented by: *Discoaster* spp., *Rhabdosphaera*
*xiphos*, *Reticulofenestra* spp. and *Gephyrocapsa* spp. (excluding *G. ericsonii*); the lower-euphotic
ecogroup contains: *Rhabdosphaera clavigera, Gephyrocapsa ericsonii* and *Ceratolithus* spp.,
finally the subeuphotic ecogroup includes: *Florisphaera profunda* and *Calciosolena murrayi*.



Because species specific stable isotope measurements and depth habitat reconstructions are
difficult for calcareous nannofossils, species depth-habitat preference was assigned based on the
literature (Poulton et al., 2017; Tangunan et al., 2018). In particular, Poulton et al. (2017) described
vertically separated coccolithophores communities sampled during a meridional cruise in the
Atlantic Ocean. Here we use their criteria for assigning nannofossil species into ecogroups,
whereby in the upper-euphotic zone ecogroup we include species found to live in waters with
>10% surface irradiance, in the lower-euphotic zone ecogroup we group species found to live in
waters with 10-1% irradiance, and in the sub-euphotic zone ecogroup we group species found to
live in waters with <1%, i.e. too low to support photosynthesis (Poulton et al., 2017). *Discoaster*
become extinct in the early Pleistocene, therefore, its depth habitat remains under debate as the
group has no extant relative (Schueth and Bralower, 2015; Tangunan et al., 2018). However,
geochemical evidence from oxygen isotope values of *Discoaster* and planktonic foraminifera
(*Globorotalia menardii*, *Dentoglobigerina altispira* and *Globigerinoides obliquus*), reveal
comparable values and signifies that *Discoaster* likely inhabited the upper euphotic zone
(Minoletti et al., 2001).
For each target age, the relative abundance of ecogroups was calculated summing up the
abundance counts of all the individual species pertaining to an ecogroup at each site, hence,
ecogroup abundance data represent global mean values. For both nannofossils and planktonic
foraminifera, the percentage of each ecogroup per time bin was converted into pie-charts (Figs. 2-
5). Diversity indexes for both foraminiferal and nannofossil ecogroup were calculated using the
statistical software Past (Hammer et al., 2001) (Fig. 6).





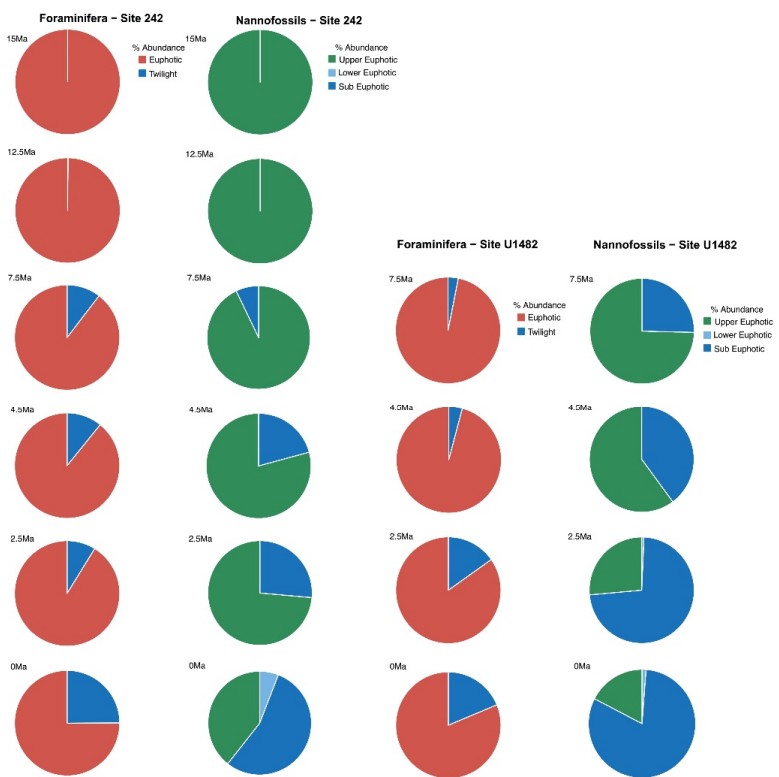


**Figure 2.** Foraminiferal and nannofossil ecogroup abundance at Site 242 and U1482.












**2.4 cGENIE model**

We extracted model output for Particulate Organic Carbon (POC) and oxygen concentration from the cGENIE simulations for each of the seven target ages as described fully in Boscolo-Galazzo, Crichton et al. (2021). To facilitate a general discussion of near-surface changes, we divided the data latitudinally by calculating the arithmetic mean for low latitudes (<16° latitude), two mid-latitude bands (mid 1: 16° to 40°, mid 2: 40° to 56°) and high latitudes (>56°). The cGENIE simulations take account of changing boundary conditions including $CO_2$ forcing, bathymetry and ocean circulation (Crichton et al., 2020). The model's ocean biological carbon pump is temperature dependent, where temperature affects both nutrient uptake rates at the surface and remineralization rates of sinking particulate organic matter down the water column (Crichton et al., 2021).

**3. Results**

**3.1 Plankton Ecogroups**

For both calcareous nannoplankton and planktonic foraminifera, the variation in abundance of euphotic zone and deeper-dwelling ecogroups show global patterns recognised across sites. Additionally both group indicate a long-term directionality towards increased abundance of deep-dwelling ecotypes. Among planktonic foraminifera, the twilight zone ecogroup increases in abundance through time starting at 7.5 Ma (Fig. 6). The relative abundance of the twilight zone ecogroup in the middle Miocene is 15% and it increases to ~30% in the Holocene time slice (Fig. 6). The average abundance of the euphotic zone species ecogroup in the middle Miocene is 85% and it decreases through time until reaching 60% in the Holocene (Fig. 6). In the twilight zone ecogroup we observe an increase in the total number of species from about 1-2 species at 15 Ma, to 14 species in the Holocene (Fig. 6). In the middle Miocene this group comprised 1/6 of the total





number of species in our samples, while in the Holocene it represents almost the half. All the
diversity indexes show a late Miocene to Holocene increasing trend for the twilight zone ecogroup
(Fig. 6).

Calcareous nannofossil assemblages are dominated by the upper-euphotic ecogroup from

15 to 10 Ma at all sites (Figs. 2-5). At 7.5 Ma the sub-euphotic ecogroup first becomes a significant
component of assemblages at Indian Ocean sub-tropical Sites U1482 and to some extent Site 242,
but it is not until the 4.5 Ma time slice that the sub-euphotic ecogroup becomes a significant
component of assemblages at the majority of locations (Sites 516, 871/872, 242, U1338, U1482,
U1489; Figs. 2-5). By the Holocene time slice, coccoliths of sub-euphotic species are dominant at
most locations, except at Eastern Equatorial Pacific Site U1338 (Figs. 6 and 4). At the southern
high latitude Site 1138 there is no significant contribution from coccoliths of either lower-euphotic
or sub-euphotic species at any point, although there is no data from the Pliocene to Holocene time
slices at this location (Fig. 5). Global average compositions of calcareous nannofossil assemblages
reflect the changes noted above, with a marked and rapid decline in the relative contribution of the
upper-euphotic ecogroup, and a corresponding increase in the sub-euphotic zone ecogroup through
the Pliocene and to Holocene (Fig. 6).
**3.2 Planktonic foraminiferal deep-dwelling species: depth habitat, abundance and**
**biogeography**
**3.2.1 Hirsutellids**
The only hirsutellid species occurring in our Miocene samples is *Hirsutella scitula*. At 15 Ma this
morphospecies is common only at Site 1138 (~8%), sporadically occurs at Site 516 (<1%) and is
absent at the other investigated sites (Figs. 7). Oxygen isotopes range from 0.5 to 1.3‰ (Figs. S1-





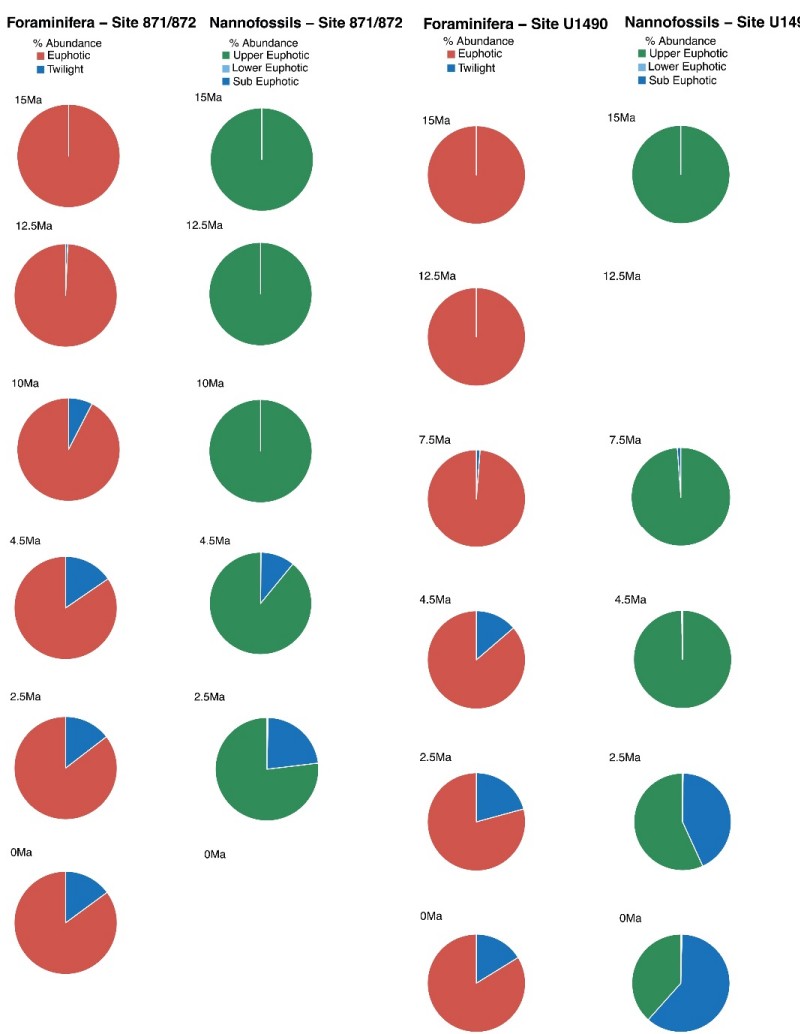



**Figure 3.** Foraminiferal and nannofossil ecogroup abundance at Site 871/872 and U1490.










S2). Depth-habitat reconstructions for Site 1138 are unattainable from $\delta^{18}O$ data due to the
overprinting effect of subpolar front shifts at this location, but habitat reconstruction at Site 516
suggests a paleodepth habitat shallower than 200 m. By 12.5 Ma, *H. scitula* appears at Site U1338
and U1489 in very low abundance (<0.5%) (Fig. 7). At Site U1489 the species was so rare that it
was encountered when picking for stable isotopes and no more when counting for species
abundances, despite the use of different splits of the residue. No differences were observed
between 12.5 and 10 Ma in the biogeography of *H. scitula* (Fig. 7). However, by 7.5 Ma, *H. scitula*
occurs at all our low latitude sites (Fig. 8) with oxygen isotopes ranging from -0.5 to 2.0‰ (Figs.
S3-S9), which according to depth habitat reconstructions translates to 250 and 500 m water depth
(Boscolo-Galazzo, Crichton et al., 2021). This is similar to that of *Globorotaloides hexagonus*
(Fig. 1), the only twilight zone dweller we observed at tropical sites at 15 Ma, displaying stable
isotopes ranging from 0 to 1‰, which translates to depths around 300-500 m. In the late middle
Miocene the stable isotope values of *G. hexagonus* increase to 2-2.5‰ (Figs. S3-S9). Similarly,
the oxygen isotope values of tropical *H. scitula* increased through time, reaching 2-3‰ in the
youngest target ages. In line with this, the reconstructed depth habitat of *H. scitula* and *G.*
*hexagonus* increases through time in a stepwise fashion, and in the Holocene it reaches down to
800-1500 m (Fig. 1) (Boscolo-Galazzo, Crichton et al., 2021). *Hirsutella scitula* becomes
gradually more common at low latitude sites through the Miocene-Pliocene, although it never
becomes abundant. In our record, *Hirsutella margaritae* and *H. theyeri* first appear in the early
Pliocene at a depth between 200-300 m (Fig. 1) (oxygen isotopes range -1 to -0.5‰), similar to
that of *H. hirsuta* (oxygen isotopes range 0 to 1‰) when it first appears in the Holocene (Figs. S3-
S9).





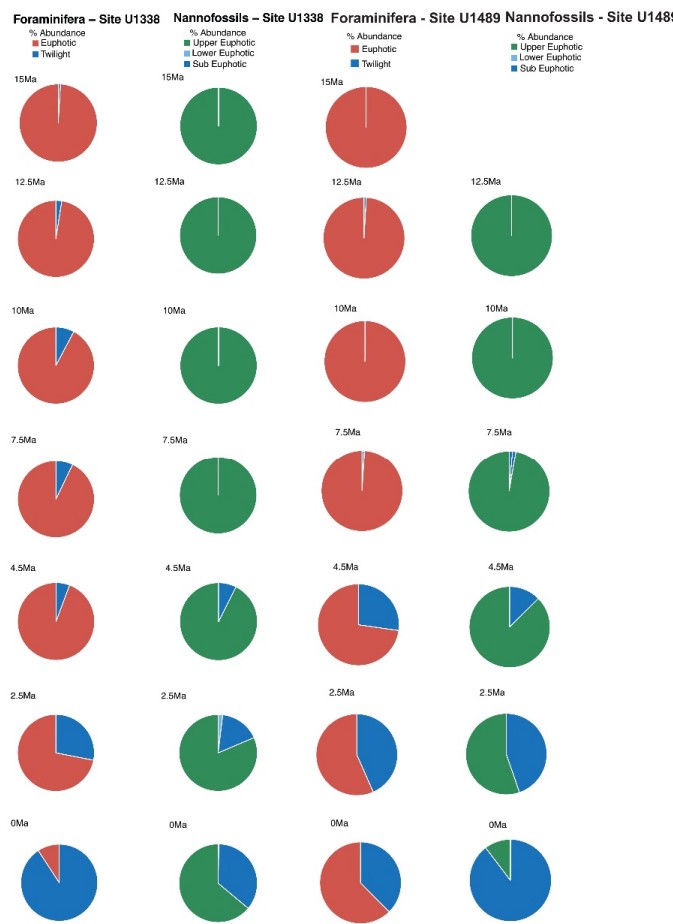




**Figure 4.** Foraminiferal and nannofossil ecogroup abundance at Site U1338 and U1489.











### 3.2.2 Truncorotaliids

The earliest appearances of *Truncorotalia crassaformis* in our record corresponds to our 4.5 Ma time slice at Site 1138 in the Indian Ocean sector of the Southern Ocean where it represents >20% of the assemblage, and in coeval sediments at mid-latitude Site 516 in the southwest Atlantic, where it represents ~9% of the assemblage (Fig. 9), with oxygen isotopes ranging from 1.5 to 3.0‰ (Figs. S1-S2). At Site 516 we observe the co-occurrence of *T. oceanica* (~14%) and *T. crassaformis* in the early Pliocene, and of *T. viola* (~5%) and *T. crassaformis* (~13%) in the late Pliocene (Fig. 10). We did not observe *T. oceanica* and *T. viola* anywhere else. *T. oceanica* and *T. crassaformis* display almost overlapping $\delta^{18}O$ values and depth habitat (Fig. S2) but *T. crassaformis* has 0.5‰ lower $\delta^{13}C$ values. Oxygen stable isotope data (1‰; Fig. S2) and habitat reconstructions for Site 516 indicate that a subsurface habitat (>200 m) was already occupied by *T. crassaformis* at the beginning of its evolutionary history (Fig. 1). The late Pliocene appearance of *T. viola* at Site 516, which differs from *T. crassaformis* in having a more convex umbilical side, a triangular outline and a subacute profile, is associated with a shift to more positive oxygen isotope values of *T. crassaformis* (1.5‰) and to slightly greater depths (Fig. 1).

We find *T. crassaformis* by the late Pliocene at our investigated tropical and subtropical sites (Figs. 10-11), with oxygen isotope values ranging from 1.0 to 2.0‰ which translate to depth habitats of 400-600 m (Boscolo-Galazzo, Crichton et al., 2021; Fig.1). The appearance of *T. crassaformis* at our low latitude sites is coeval with the appearance in our record of *Truncorotalia tosaensis*, morphologically transitional between *T. crassaformis* and *T. truncatulinoides* (Lazarus et al., 1995) (Fig. 10). *Truncorotalia tosaensis* displays oxygen isotopes values ranging from 0 to 0.5‰ (Figs. S1-S9), which translate to 300-350 m depth (Fig. 1). Consistent with earlier findings (Lazarus et al., 1995), we record the first occurrence of *T. truncatulinoides* in the late Pliocene in





the south-west Pacific (Site U1482), and only later in the North Pacific (Site 872), Indian Ocean
(Site 242) and South Atlantic (Site 516) (Fig. 11). *Truncorotalia truncatulinoides* records oxygen
isotope values ranging from -1 to 2‰, more negative than coeval *T. crassaformis* (Figs. S3-S9).
*Truncorotalia truncatulinoides*, although reported in the modern tropical ocean as one of the
species living at the greatest depths, occupies a shallower depth habitat than *T. crassaformis* when
it first appears in our tropical to subtropical records (2.5 Ma).





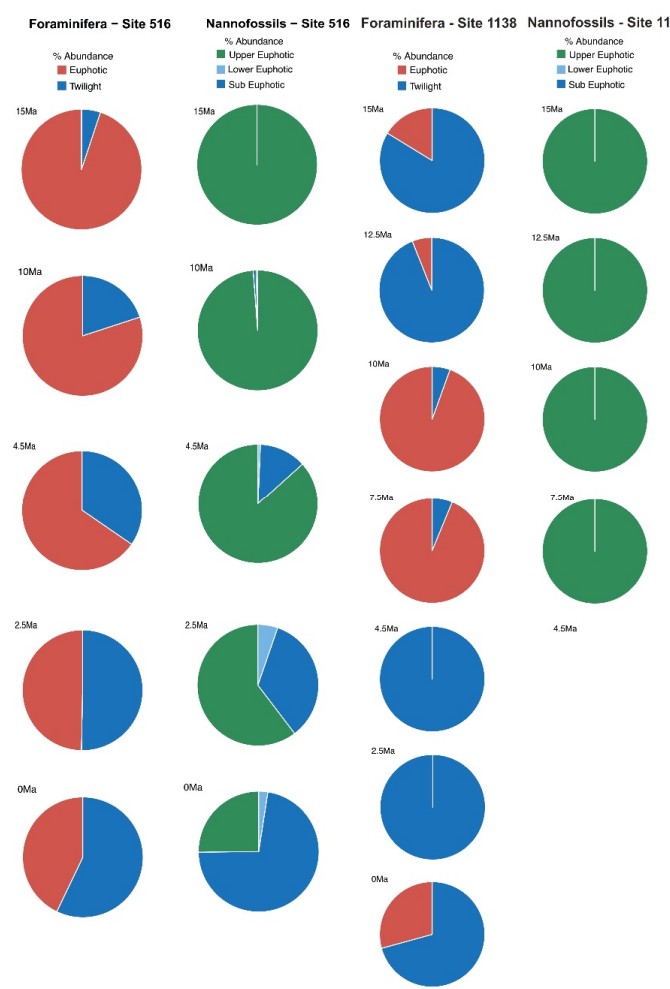


**Figure 5.** Foraminiferal and nannofossil ecogroup abundance at Site 516 and 1138.






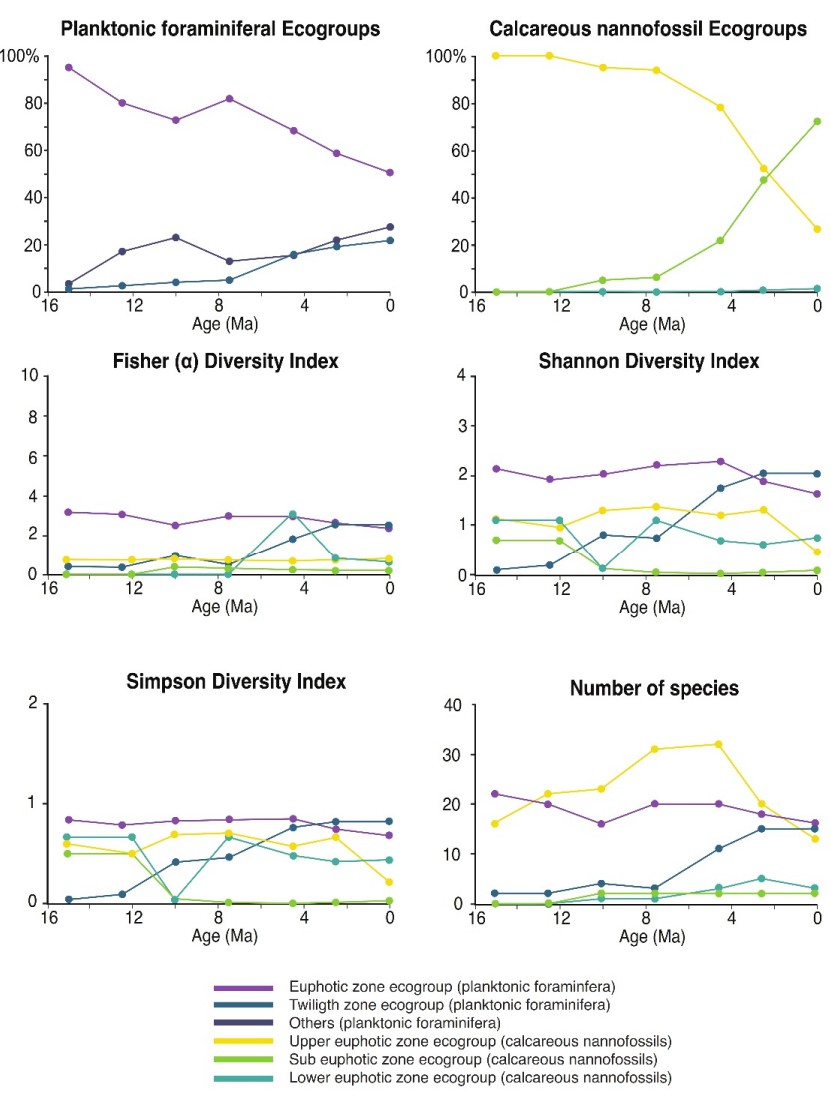


**Figure 6.** Planktonic foraminiferal and nannofossil ecogroups relative abundance and diversity indexes.











### 3.2.3 Globorotaliids


With globorotaliids here we refer to the *Globorotalia merotumida-tumida* lineage composed by *G.*
*merotumida*, *G. pleosiotumida*, *G. tumida* and *G. ungulata* (Kennett and Srinivasan, 1983). This
group first appears with *Globorotalia plesiotumida* in our 10 Ma time slice at Site 871 (Fig. 7). At
all the investigated low latitude sites, we find *G. tumida* by 4.5 Ma with abundances between 2-
24% (Fig. 9). *Globorotalia plesiotumida* co-occurs with *G. tumida* only at Sites 872 and 242
corresponding to our 4.5 Ma time slice (Fig. 9; Fig. S9). In our records, *G. tumida* consistently
displays oxygen isotope values between -1 to 0‰ (Figs. S3-S9), and an average depth habitat
around 250 m, with a shallowest occurrence at 50 m and a deepest occurrence around 600 m (Fig.
1). Similar oxygen isotope values and depth habitat preference are recorded for *G. plesiotumida*
and *G. ungulata* (Fig. 1 and Figs. S3-S9).

### 3.2.4 Globoconellids


In our records, *Globoconella miozea* is a dominant component of planktonic foraminiferal
assemblage at Site 1138 at 15 Ma (65%) and occurs in moderate abundance at Site 516 (5%) (Fig.
7). The distribution of globoconellids appears restricted to southern high to mid-latitudes during
the middle Miocene and the late Miocene to Pliocene. At Site 1138 globoconellids decrease in
abundance through time, with *Globoconella panda* the only late Miocene species (<1%; with $\delta^{18}O$
of 3‰), followed only by *Globoconella inflata* in the Holocene (4.2%; with $\delta^{18}O$ of 3.5‰) (Fig.
11). On the contrary, at Site 516 globoconellids increase in abundance through time becoming a
characteristic feature of the planktonic foraminifera assemblage as noted in previous studies of this
area (Berggren, 1977; Norris et al., 1994) (Figs. 7-11). In the Holocene *G. inflata* is most abundant
at mid-latitude Site 516 (22.7%; with $\delta^{18}O$ of 1‰), but also occurs in the subtropical (<0.5%; with
$\delta^{18}O$ of 0.9‰) and subpolar Indian ocean (4.2%; with $\delta^{18}O$ of 3.5‰) (Fig. 11; Figs. S1-S3). Depth



habitat reconstructions for the globoconellids show a deepening trend through time although less
marked compared to those of other deep-dwelling groups considered in this study (Boscolo-
Galazzo, Crichton et al., 2021; Fig. 1). At Site 516, the depth habitat is just above 200 m for middle
Miocene *Globoconella miozea* ($\delta^{18}$O 0.9‰), just below 200 m for late Miocene *G. conoidea* and
*G. conomiozea* ($\delta^{18}$O 1‰), and around 350 m for late Pliocene *Globoconella puncticulata* ($\delta^{18}$O
1.6‰) the precursor of *G. inflata,* which shows a similar depth habitat at this site ($\delta^{18}$O 1-1.2‰;
Fig. S2), (Fig. 1). At Site 242 the average depth reconstruction for the Holocene *G. inflata* is of
~450 m ($\delta^{18}$O 0.9‰; Fig. S3), similar to that of *T. crassaformis* and *T. truncatulinoides* (Boscolo-
Galazzo, Crichton et al., 2021; Fig. 1).
**4. Discussion**
**4.1 Evolution of a deep-plankton ecological niche linked to late Neogene cooling**

We observe a trend of  increasing ecological importance of deep-dwelling species in both

calcareous nannofossil and foraminiferal communities from the late Miocene to Recent. The
foraminiferal twilight zone ecogroup shows a marked increase in abundance and diversity at 7.5
Ma, at the same time as the first significant appearance of the sub-euphotic ecogroup within
coccolithophore assemblages, and coinciding with a possible acceleration of global cooling
(Cramer et al., 2011; Crichton et al., 2020). Fossil deep-dwelling coccoliths are dominated by one
species, *Florisphaera profunda,* which is often very abundant in Plio-Pleistocene sediments. This
makes our sub-euphotic ecogroup a low diversity – high abundance assemblage, whose origin
significantly impacts the relative abundance balance between ecogroups, but with little change in
diversity metrics towards the modern. Although there are morphological variants of *F. profunda*
in the modern oceans, potentially representing sub-species or pseudo-cryptic species (Quinn et al.,
2005), these are not distinguished in fossil assemblages and documenting their divergence times





requires either further molecular genetic or detailed morphological analyses. The one clear signal
in our diversity analyses is a late Miocene – early Pliocene peak in upper-euphotic species richness,
followed by a marked decline through the late Pliocene to the Holocene. The late Miocene-
Pliocene peak diversity is present in previous global compilations of total nannofossil diversity
(Bown et al. 2004; Lowery et al. 2020), but here we show that this signal is driven by first a
diversification and then progressive extinction almost entirely within upper-euphotic taxa.
Modern planktonic foraminifera evolved in two main diversification pulses in the middle Miocene
(16-14 Ma) and during the late Miocene-Pliocene transition (6-4.5 Ma) (Kennett and Srinivasan,
1983; Kucera and Schönfeld, 2007). Our species abundance and diversity data show that this
diversification was mostly driven by the origin of lineages of deep/subsurface dwelling species
(Figs. 7-11 and Fig. 6). Diversity among euphotic zone species remained constant from the middle
Miocene to the early Pliocene, then declined (Fig. 6). This pattern, similar to calcareous
nannofossils, may explain early records pointing to a decrease in Pliocene to Recent planktonic
foraminiferal diversity (Wei and Kennett, 1986).

The observed evolutionary patterns in planktonic foraminifera and coccolithophores can

be explained by the development of environmental conditions favourable to deep living plankton



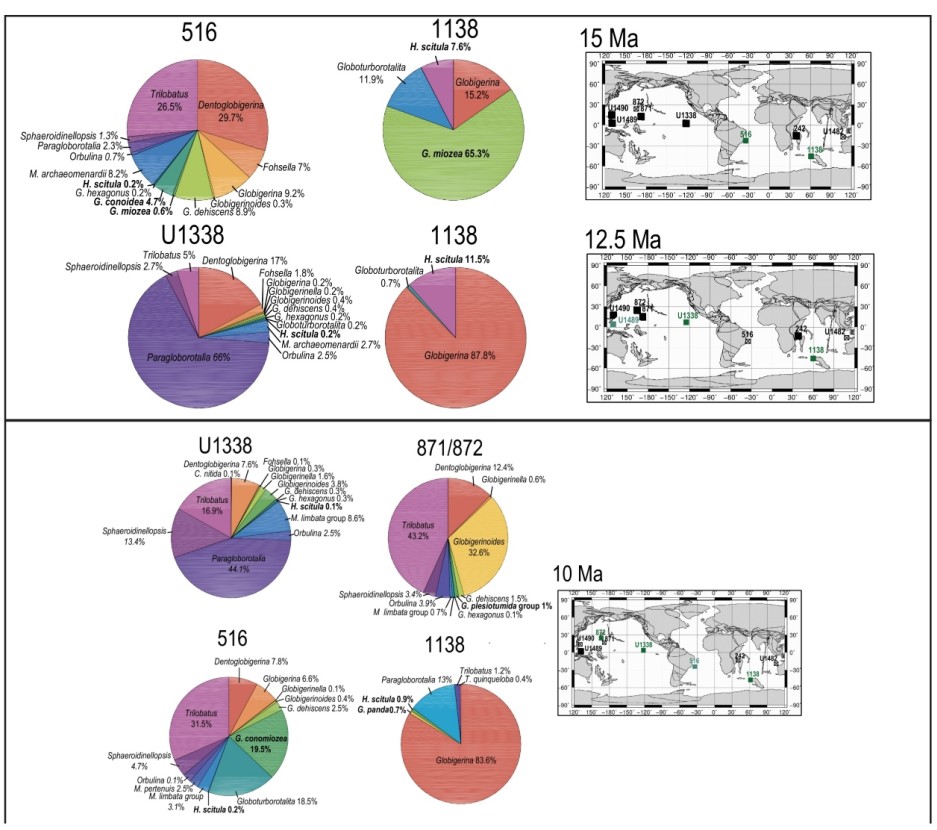

**Figure 7.** Abundance and biogeography of planktonic foraminiferal species at 15, 12.5, and 10

Ma. In the pie-charts twilight zone planktonic foraminiferal species are in bold. Sites where

twilight zone planktonic foraminifera were found are highlighted in green in the maps. The crossed

square symbol in the maps indicate that the time interval of interest was not recovered for a given

site. Continental configuration follows: Ocean Drilling Stratigraphic Network (GEOMAR, Kiel,

Germany).



with cooling. For the deep-dwelling planktonic foraminifera, survival requires the availability of
food at depth, and with the exception of few species adapted to oxygen minimum zones (Davis et
al., 2021), the absence of severe oxygen depletion. In the published literature there is a general
tendency for planktonic foraminiferal $\delta^{18}O$ values to be tightly grouped during times of warm
climate, becoming more spread-out during cooling episodes, for instance the transition from mid-
to late Cretaceous (Ando et al., 2010) and early to middle Eocene (John et al., 2013). Because our
data are restricted to certain time slices and locations, we only capture a part of the overall pattern
for the Neogene, but other examples of depth-related evolution in the Neogene include the
*Fohsella peripheroronda – F. fohsi* lineage in the middle Miocene (Hodell and Vayavananda,
1993; Norris et al., 1993) and the appearance of various deep-dwelling digitate species in the Plio-
Pleistocene (Coxall et al., 2007).
Deep-dwelling coccolithophores, most notably *Florisphaera profunda*, require dissolved
macronutrients (N, P) and at least some degree of light penetration (Quinn et al., 2005; Poulton et
al., 2017). The requirement for light penetration to depth is most clearly shown by the absence
of *F. profunda* beneath high surface productivity regions where nutrients are not limited at depth
but light is rapidly attenuated by more abundant mixed layer microplankton (Beaufort et al., 1999).
For coccolithophores, the cooling-driven shift from nutrient recycling within to below the mixed
layer, may have provided the ecological driver for species to live in deeper, more nutrient-rich
waters, but, as mixed layer waters cleared, also allowed the irradiance necessary for photosynthesis
to penetrate to these new deeper habitats. Additionally, the capability of coccolithophores to absorb
carbon and nutrients from seawater under low light conditions (Godrjian et al., 2020) may have
also aided in the occupation of new deep water niches.



Our model outputs from cGENIE is consistent with this interpretation, organic matter
export (POC at 40 m in Fig. 12) reduced with cooling, suggesting an overall decrease in primary
productivity at all 4 latitudinal bands considered in this study. Fewer particles in surface waters
would have allowed greater light penetration (Fig. 12), at the same time, the model indicates
enhanced organic matter delivery at >200 m with cooling.  Greater organic matter delivery below
the euphotic zone, suggests a deeper remineralization depth and increased dissolved nutrients
availability at depth. This is most clearly shown in the modelled low and mid latitudes near-surface
waters. Oxygen availability also increased, particularly below 100 m depth in low latitudes and
further down the water column, below 200 m depth (Fig. 12).
Nonetheless, planktonic foraminifera and nannoplankton have distinct trophic statuses
(zoo- versus phytoplankton), further coccolithophore species require light for their dominantly
photosynthetic mode of life. In our data, such differences between the requirements of zoo- and
phytoplankton deep-dwellers is clearly observed in the biogeographic patterns. Sub-euphotic
coccolithophores are consistently more abundant in low nutrient sub-tropical locations (e.g. DSDP
Site 242 and IODP Site U1482; Fig. 2). The end-member of this biogeographic difference is ODP
Site 1138, in the Southern Ocean. Here, twilight foraminifera dominate most time intervals,
presumably due to high export production. However, lower-euphotic and sub-euphotic
coccolithophores are effectively excluded, due to turbid, high-productivity surface waters (Fig. 5).
This pattern is also supported through a comparison of high to low productivity tropical sites in
the Holocene – the Eastern Equatorial Pacific Site U1338 has abundant twilight foraminifera, but
relatively low abundances of sub-euphotic taxa (Fig. 4), whereas the more oligotrophic Site U1482
(Fig. 2) has lower abundances of twilight foraminifera and higher abundances of sub-euphotic
coccolithophores.





Despite these ecological differences between zoo- and phytoplankton, there is a shared

environmental driver for the evolution of deep-dwelling coccolithophores and planktonic
foraminifera linking the evolution of deep-dwelling specialists in each group. Efficient near-
surface recycling of organic carbon in past warm climates, such as the middle Miocene, precluded
the occupation of the deep habitat for both groups by reducing both organic carbon transfer (food
limitation for foraminifera and for foraminiferal prays such copepods) and light penetration
(photosynthesis for coccolithophores) to depth. Global cooling since the middle Miocene (Kennett
and Von der Borch, 1985; Kennett and Exon, 2004; Cramer et al., 2011; Zhang et al., 2014; Herbert
et al., 2016; Sosdian et al., 2018; Super et al., 2020), however, led to a decreased export of organic
matter and a deepening of the mean organic matter remineralization depth, which in turn favoured
the evolution of deep water niches in planktonic foraminifera and nannoplankton via increased
availability of organic matter, oxygen, and likely nutrients at depth, and clearing of surface waters.

**4.2 Mechanisms of speciation of deep-dwelling planktonic foraminifera**

The hirsutellids gave rise to a large late Neogene radiation among planktonic foraminifera,

leading to the origin of modern phyletic groups such as the menardellids, globoconellids,
truncorotaliids, and the globorotaliids of the *Globorotalia merotumida - tumida* lineage (Kennett
and Srinivasan, 1983; Aze et al., 2011). The majority of the modern representatives of these groups
are lower euphotic zone to twilight zone species. The hirsutellids originated about 18 Ma (Wade
et al., 2011) from *Globorotalia zealandica* (Kennett and Srinivasan, 1983), the first representative
of the group being *Hirsutella praescitula* (Kennett and Srinivasan, 1983; Aze et al., 2011). Extant
hirsutellids include *H. scitula*, *H. hirsuta* and *H. theyeri*; genetic data available for *H. hirsuta*
indicate a single genotype (Schiebel and Hembleben, 2017). Modern *H. scitula* and *H. hirsuta* are



deep water forms. Depth habitat reconstructions show *H. scitula* at the greatest water depth
(Boscolo-Galazzo, Crichton et al., 2021; Fig. 1), consistent with it being reported to have a deeper
average depth habitat than *H. hirsuta* in the modern ocean (Birch et al., 2013; Stainbank et al.,
2019), where it feeds on suspended organic matter (Itou et al., 2001). *Hirsutella hirsuta* has been
reported to feed on dead diatoms and to predominantly live at depths around 250 m (Schiebel and
Hemleben, 2017), consistent with our habitat reconstructions placing *H. hirsuta* and other
hirsutellids shallower than *H. scitula*, between the bottom of the euphotic zone and the upper
twilight zone. Our depth habitat reconstruction for *H. scitula* at Site 516 for the 15 Ma time slice,
indicates an initial euphotic zone depth habitat preference for this species. By the 7.5 Ma time
slice, we find *H. scitula* at most of our low latitude sites, at depth comprised between 300-500 m
(Fig. 1). We suggest that the spread of *H. scitula* from high-mid latitudes towards the tropics after
the middle Miocene warmth (Figs. 7-11) tracks increasing availability of POC and oxygen at depth
(Boscolo-Galazzo, Crichton et al., 2021; Fig. 12), allowing this species to find food in deep tropical
twilight zone waters, profiting from a new ecological niche. We suggest that moving from a surface
to a deep-water habitat was for *H. scitula* an ecological innovation which allowed the species to
move outside its high-mid latitude areal, consistent with observations from the modern ocean
documenting *H. scitula* dwelling at progressively deeper depth from high to tropical latitudes
(Schiebel and Hemleben, 2017). The proceeding ocean cooling (and increasing efficiency of the
biological pump) explains the stepwise depth habitat increase of *H. scitula* through time at tropical
and subtropical sites (Fig.1): more food became available at increasingly greater depth in
association with improved oxygen availability (Fig. 12), allowing the species to expand its habitat



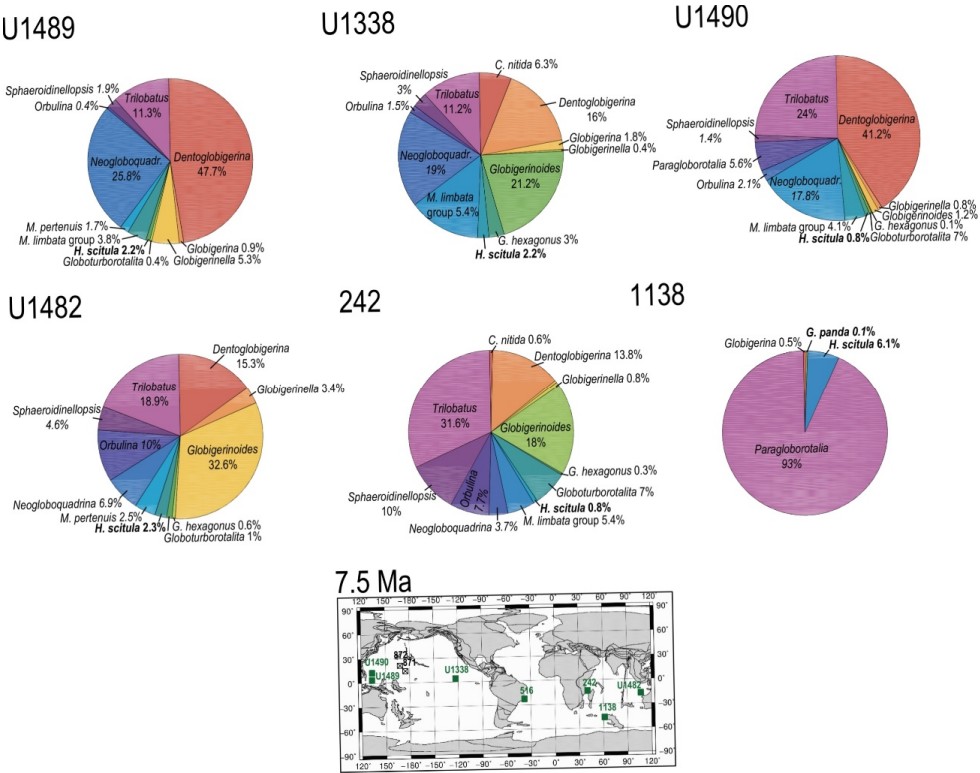

**Figure 8.** Abundance and biogeography of planktonic foraminiferal species at 7.5 Ma. In the pie-charts twilight zone planktonic foraminiferal species are in bold. Sites where twilight zone planktonic foraminifera were found are highlighted in green in the maps. The crossed square symbol in the maps indicate that the time interval of interest was not recovered for a given site. Continental configuration follows: Ocean Drilling Stratigraphic Network (GEOMAR, Kiel, Germany).



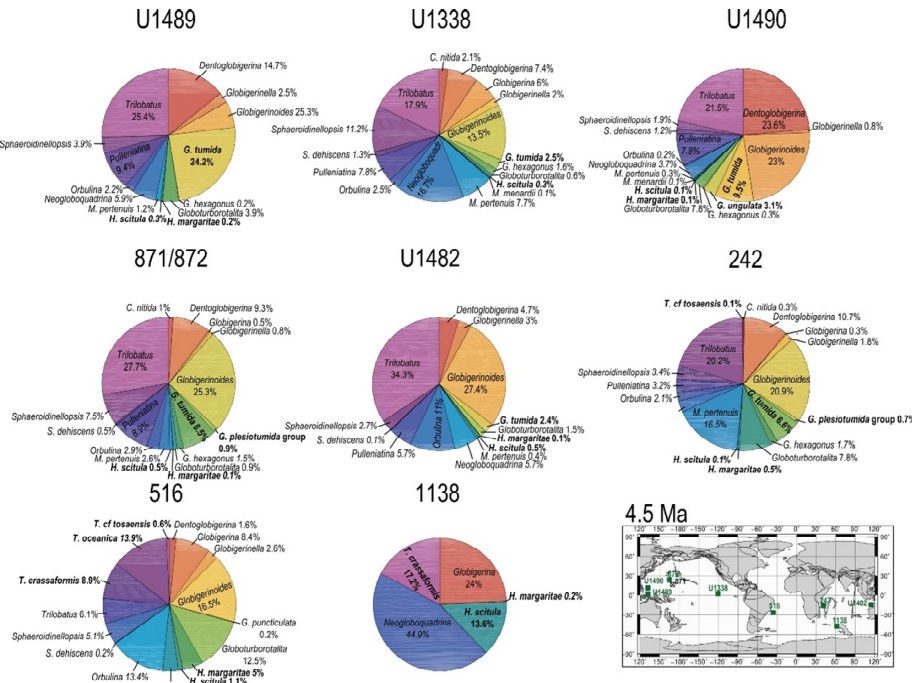

**Figure 9.** Abundance and biogeography of planktonic foraminiferal species at 4.5 Ma. In the pie-charts twilight zone planktonic foraminiferal species are in bold. Sites where twilight zone planktonic foraminifera were found are highlighted in green in the maps. The crossed square symbol in the maps indicate that the time interval of interest was not recovered for a given site. Continental configuration follows: Ocean Drilling Stratigraphic Network (GEOMAR, Kiel, Germany).





vertically other than geographically. At low latitude sites this was accompanied by speciation, with
the appearance of *Hirsutella margaritae* and *H. theyeri* in the early Pliocene, and of *H. hirsuta* in
the Holocene. These new species display depth habitats shallower than *H. scitula* at their
appearance. Hence, we suggest that it was the occupation of a new deep water habitat in tropical
waters that triggered speciation from *H. scitula* through depth sympatry, i.e. genetic isolation
attained in the same area but at different depths (Weiner et al., 2012). Earlier studies on speciation
among planktonic foraminifera based on the fossil record, highlighted a predominance of
sympatric speciation, possibly linked to changes in ecology (Norris et al., 1993; Lazarus et al.,
1995; Pearson et al., 1997). This has more recently been supported by genetic analysis, which
reveals a consistent depth separation between intra-specific genotypes at a global scale, suggesting
that depth sympatry could be a universal mechanism generating diversity among microplankton
(Weiner et al., 2012).
Truncorotaliids start their evolutionary history at ~4.31 Ma (Raffi et al., 2020), when
*Truncorotalia crassaformis* splits from *Hirsutella cibaoensis* (Kennett and Srinivasan, 1983; Aze
et al., 2011), one of the first new species originating from *H. scitula* after its spread to low latitudes
(Kennett and Srinivasan, 1983). Our reconstructed geographic and temporal distribution for *T.*
*crassaformis* suggests that the early Pliocene split from the hirsutellids happened in cold subpolar
water. By the early Pliocene *Truncorotalia crassaformis* was an abundant component of the
subpolar assemblage at Site 1138 (>17%) and had already successfully spread to middle latitudes
(>8%) but does not occur in our low latitude samples (Fig.9).






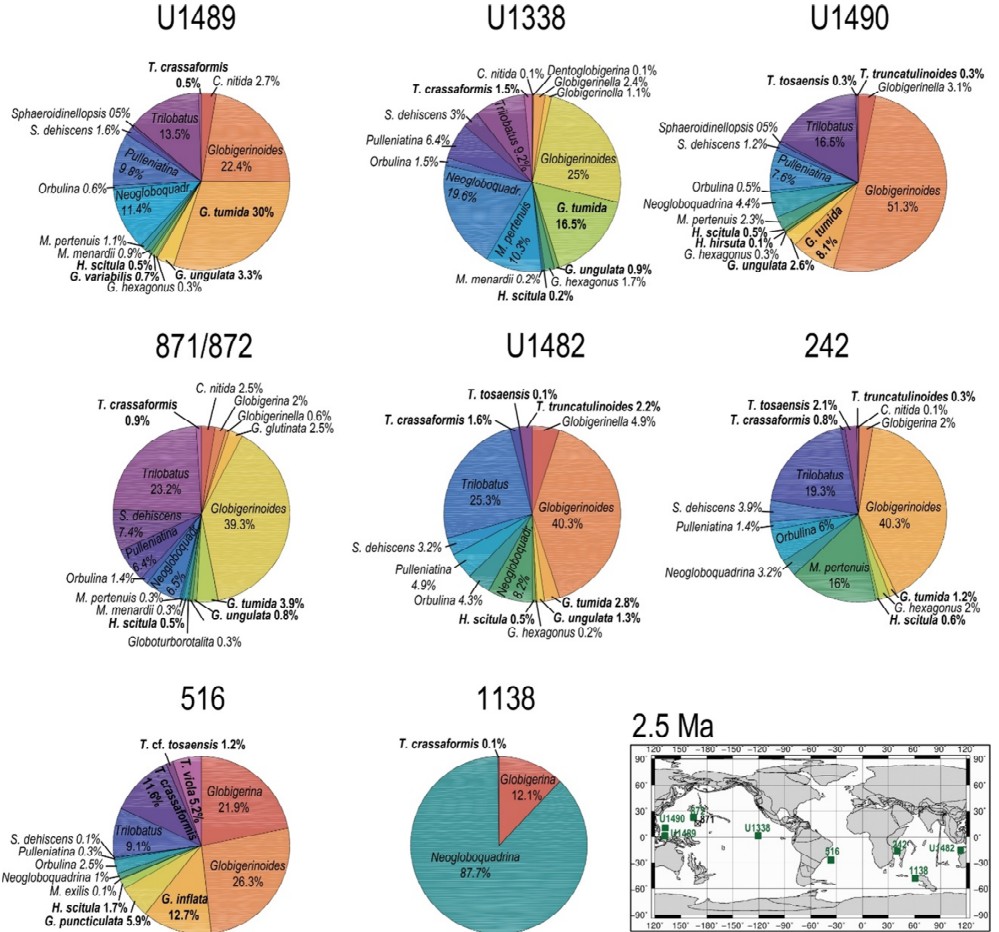


**Figure 10.** Abundance and biogeography of planktonic foraminiferal species at 2.5 Ma. In the pie-charts twilight zone planktonic foraminiferal species are in bold. Sites where twilight zone planktonic foraminifera were found are highlighted in green in the maps. The crossed square symbol in the maps indicate that the time interval of interest was not recovered for a given site. Continental configuration follows: Ocean Drilling Stratigraphic Network (GEOMAR, Kiel, Germany).



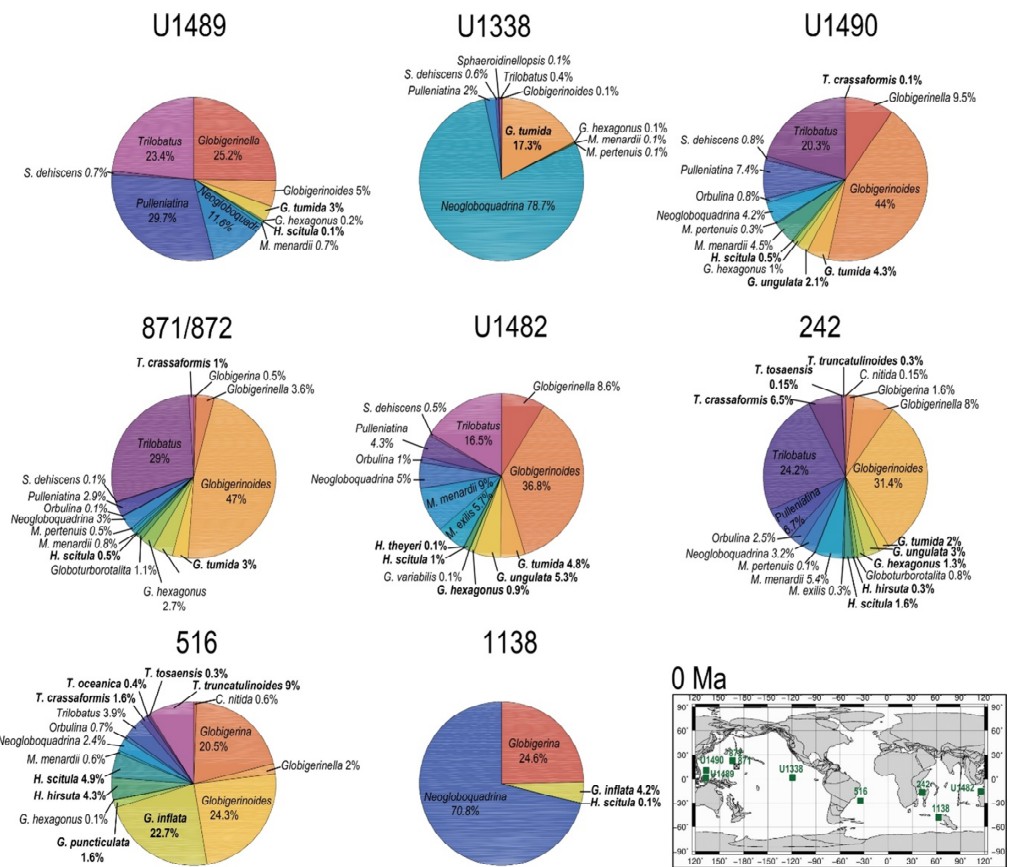

**Figure 11.** Abundance and biogeography of twilight zone planktonic foraminiferal species at 0 Ma. In the pie-charts twilight zone planktonic foraminiferal species are in bold. Sites where twilight zone planktonic foraminifera were found are highlighted in green in the maps. The crossed square symbol in the maps indicate that the time interval of interest was not recovered for a given site. Continental configuration follows: Ocean Drilling Stratigraphic Network (GEOMAR, Kiel, Germany).



We suggest that the evolution of *Truncorotalia crassaformis* from the hirsutellids may have
happened through allopatry, with low latitude population of *H. cibaoensis* becoming isolated from
subpolar populations and eventually evolving into *T. crassaformis*. Similar to *H. scitula,* from
subpolar latitudes, *Truncorotalia crassaformis* appears to have subsequently spread to lower
latitudes occupying progressively deeper habitats (Fig. 1; Figs. 7-11), and originating numerous
daughter species, some of which are intermediate morphospecies with limited geographic
distributions no longer extant today (Lazarus et al., 1995). At Site 516 we find *Truncorotalia viola*,
with lighter $\delta^{18}$O and $\delta^{13}$C values than *T. crassaformis* (Fig. S2), pointing to a clear ecological
differentiation. Together with the marked morphological differentiation between the two, this
suggests *T. viola* may have been a different biological species. *Truncorotalia truncatulinoides*, the
most representative species of this group, appears to have originated from *T. crassaformis* at about
2.7 Ma in the tropical southwest Pacific, and subsequently spread in the global ocean (Dowsett,
1988; Lazarus et al., 1995). According to our reconstruction, depth sympatry associated with
gradual morphological changes characterizes speciation among the truncorotaliids, as depth habitat
deepening of the ancestor/precursor is clear in the transition *T. crassaformis-T. viola* and *T.*
*crassaformis-T. tosaensis-T. truncatulinoides* in our record (Fig. 1). The possibility to colonize
deeper water habitats may have led to progressive reproductive isolation between "deeper" and
"shallower" populations of *T. crassaformis*, resulting in biological speciation. Depth sympatry as
speciation mechanism for the truncorotaliids was already proposed by Lazarus et al. (1995) based
on biogeography, but without a definitive test.

In the modern ocean the *Globorotalia merotumida-tumida* lineage is represented by *G.*

*tumida* and *G. ungulata*, distributed in tropical to temperate regions. Genetic analysis has revealed
that they are two distinct biological species with a single genotype each (Schiebel and Hemleben,



2017). Little is known about the ecology of these two species, although *G. tumida* is known to
dwell in subsurface waters at the deep chlorophyll maximum (Schiebel and Hemleben, 2017).
According to the phylogeny of Aze et al. (2011), *Menardella menardii* gave rise to the
*Globorotalia merotumida-tumida* lineage around 9 Ma. However, *Menardella menardii* is absent
in our late Miocene samples while other menardellids such *Menardella limbata* and *M. pertenuis*
are common at several of our investigated middle latitudes to tropical sites. By 4.5 Ma,
*Globorotalia tumida* had evolved and is common at all of our low latitude sites (Fig. 9), while *M.*
*menardii* is extremely rare, occurring only at Sites U1338 and U1490 (0.1%), becoming more
common only by the Holocene. Based on this biogeographic pattern, we propose that the *G.*
*merotumida-tumida* lineage originated from a late Neogene menardellid, such for instance *M.*
*limbata* which is morphologically very similar to *G. merotumida*. Given that *G. plesiotumida* and
*G. tumida* display a deeper habitat (> 200 m) than *M. limbata* and other Miocene menardellids
(100-200 m), we suggest that such transition may have happened through depth sympatry in
tropical waters, with forms which remained reproductively isolated in the twilight zone generating
the *G. merotumida-tumida* lineage. Morphometric measurements on *M. limbata* and *G.*
*merotumida* shells are required to test for an evolutionary relationship between these two species.
According to our depth habitat reconstruction, *G. plesitumida* and *G. tumida* occupied a similar
depth habitat at 4.5 Ma, so it is not clear from our dataset which evolutionary mechanism may
have led to the origination of the latter from the first. Hull and Norris (2009) analyzed speciation
within this lineage and suggested that the evolution from *G. plesiotumida* to *G. tumida* happened
abruptly within 44 kyr. *Globorotalia ungulata*, appears in our record by the late Pliocene and often
display a habitat shallower than *G. tumida*, suggesting depth sympatry as the evolutionary
mechanism leading from *G. tumida* to *G. ungulata*. However, because depth habitat



reconstructions for these two species are more variable and shallower than that of other twilight
zone species, more data are required to more conclusively infer speciation mechanisms.

The globoconellids originated in the late early Miocene (~17 Ma) with *Globoconella*

*miozea*, which is considered to descend directly from *Hirsutella praescitula* (Kennett and
Srinivasan, 1983; Norris et al., 1994; Aze et al., 2011). *Globoconella conoidea* originated in the
middle Miocene and, after the extinction of *G. miozea* at about 10 Ma, remained the only
representative of the globoconellids until the latest Miocene. At this time, the evolutionary
turnover within the group accelerated and a number of different morphospecies originate from *G.*
*conoidea* and go extinct very rapidly, until the appearance in the late Pliocene of *G. inflata* which
persists until today (Wei and Kennett, 1988; Wei, 1994; Aze et al., 2011). Compared to other
Neogene to Recent taxa, the globoconellids display a more restricted geographic distribution
throughout their evolutionary history. They tend to be common at mid latitude hydrographic fronts
(Schneider and Kennett, 1999; Schiebel and Hemleben, 2017; Lam and Leckie, 2020), except *G.*
*puncticulata* and *G. inflata*, which extend into low latitude regions (Norris et a., 1994). The
geographic distribution of globoconellids as shown here suggests that this group was already
specialized to live at hydrographic fronts in the middle Miocene, possibly feeding on
phytoplankton. Starting at ~5.5 Ma, cooling and the possibility to feed on sinking detritus in deeper
waters (Boscolo-Galazzo, Crichton et al., 2021) may have stimulated evolutionary turnover within
this otherwise rather static group. The closely spaced temporal succession of morphospecies at this
time may reflect ongoing evolutionary experiments in an attempt to profit from new ecological
possibilities opening up at depth and outside the area of the group. The depth habitat
reconstructions for *G. puncticulata* and *G. inflata* suggests that from the Pliocene this group started



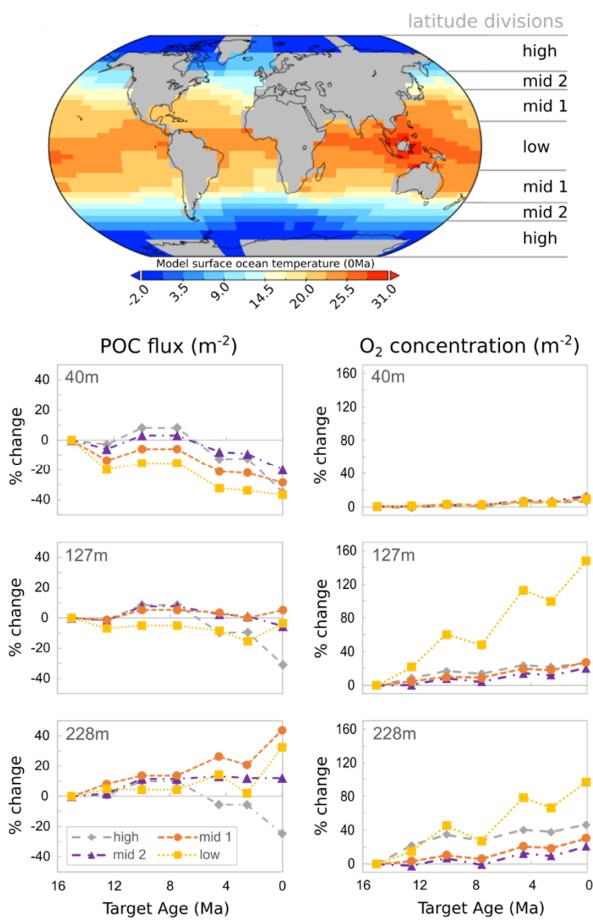


**Figure 12.** cGENIE model output for changes in Particulate Organic Carbon (POC) flux, and

dissolved Oxygen in near-surface ocean waters from the middle Miocene to Present, with a

temperature-dependent biological carbon pump. Inset map shows the modelled Present surface

ocean temperatures. Depths are the middle of cGENIE's top three ocean layers.




to progressively adapt to greater depths, consistent with the distinctive change in morphology
between G. sphericomiozea (and other Miocene globoconellids) and its Pliocene descendants *G.*
*puncticulata* and *G. inflata* (Kennett and Srinivasan, 1983). We suggest that an evolutionary
transition began with the morphospecies *G. puncticulata*, transitional between *G. sphericomiozea*
and *G. inflata* and led to the late Pliocene speciation of *G. inflata*. It is not clear from our data
whether depth sympatry or allopatry allowed the speciation of *G. inflata,* as *G. puncticulata* and
*G. inflata* show similar depth habitat in our record.  It may have been a combination of the two,
given *G. inflata* genotypes display a characteristic allopatric distribution in the ocean (Morard et
al., 2011).
Our data indicate that combining stable isotopes and model-derived water column temperature
is a promising approach to quantify the depth habitat of extinct planktonic foraminiferal species.
When combined with abundance and biogeographic data, depth habitat reconstructions offer
insights into speciation mechanisms not resolvable with the use of one technique alone (e.g., stable
isotopes). Our reconstructions indicate that both allopatry and depth sympatry played a role in the
origin of modern deep-dwelling planktonic foraminiferal species. Both allopatry and depth
sympatry appear to have been involved with the cladogenesis of the truncorotaliid and the
globorotaliid lineages, while depth sympatry seems to be mostly involved for intra-lineage
speciation.
**5. Conclusions**
Our global abundance and biogeographic data combined with our depth habitat
reconstructions allow us to piece together the environmental drivers behind speciation in two of
the most extensively studied group of pelagic microfossils, planktonic foraminifera and calcareous
nannofossils over the last 15 million years. The evolution of the new Neogene deep-water lineages





of the hirsutellids, globorotaliids, globoconellids and truncorotaliids, and of nannoplankton lower-
euphotic zone and sub-euphotic zone species, resulted in the vertical stratification of species seen
in the modern ocean, in particular at low latitudes. For planktonic foraminifera such vertical
stratification of species, hundreds of meters below the surface, originated through depth sympatry
as well as cladogenesis of new lineages via both sympatry and allopatry.
Our study places the evolution of modern plankton groups in the context of large scale
changes in ocean macroecology driven by the global climate dynamics of the past fifteen million
years. The late Miocene to present evolutionary history of planktonic foraminifera and
nannoplankton was linked, wherein increased efficiency of the biological pump with cooling since
the middle Miocene was a shared evolutionary driver. Lower rates of organic matter
remineralization in the upper part of the water column allowed the creation of new ecological
niches in deep waters, by increasing food delivery and oxygen at depth for heterotrophic planktonic
foraminifera, and by clearing surface waters and augmenting the concentration of macronutrients
at depth for nannoplankton.



**Data availability:** All data associated with this article are available in the Supplement or have
been previously published. The code is tagged as v0.9.18 and is available at DOI:
10.5281/zenodo.4469673 and at: DOI: 10.5281/zenodo.4469678.

**Author contributions:**
Conceptualization, F.B.G. and P.N.P.; investigation, F.B.G. (planktonic foraminifera) and A.J.
(nannofossils); software, K.A.C.; writing–original draft, F.B.G.; writing–review and editing,
F.B.G., A.J., T.D.J, K.A.C., P.N.P., B.S.W.; visualization, F.B.G., A.J., K.A.C.; project
administration and funding acquisition, P.N.P. and B.S.W.

**Competing interests:**
The authors declare that they have no conflict of interest.
**Acknowledgements:**
Supported by Natural Environment Research Council (NERC) grant NE/N001621/1 to P.N.P.
(F.B.G. and K.A.C.), NERC grant NE/P019013/1 to B.W and NERC grant NE/P016375/1 to
participate in IODP Expedition 363 (P.N.P.). A.J. was founded by a CENTA Doctoral Training
Programme as part of NERC grant: NE/L002493/1 to T.D.J. We thank Ian Hall for becoming
acting Principal Investigator in the later stages of the grant and Jamie Wilson for insightful
discussions.

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
