# Peer review of "Late Neogene evolution of modern deep-dwelling plankton"

_Biogeosciences, 2021_

## Author Comment (AC1)

This is a truly excellent paper. It represents a major advance in the understanding of the evolution of Neogene planktonic foraminifera and calcareous nanoplankton over the last 15 million years. It usefully explains, for the first time, the evolutionary diversity increases in numerous lineages that lead to the modern depth and oceanic distributions of these important planktonic groups. These groups are an important component of the modern ocean plankton and play a significant role in marine biogenic productivity. The paper is well written and easy and enjoyable to read. It builds upon the earlier (2021) paper by Boscolo-Galazzo et al. Essentially the paper demonstrates, using large new suite of quantitative taxonomic and geochemical data, that the deep-dwelling planktonic communities evolved as an important component of oceanic planktonic during the later Neogene (after about 7.5 my). This, in turn, played a vital role in increased planktonic biomass production and its expanded depth distribution in the oceans and related processes of vertical organic carbon transfer via marine snow.

The paper also convincingly argues that the development of the deep planktonic ecological niches (i.e. new deep-water ecogroups of planktonic foraminifera and calcareous nanoplankton) were driven by palaeoceanographic processes related to late Neogene oceanic (global) cooling. This sequential cooling stimulated oceanic circulation that increased organic carbon (nutrients) export production to the deep sea, increased oceanic oxygen levels, and enhanced light penetration into deeper waters, all of which stimulated the vertical biological pump that favored the evolution of these deep-water niches.

The paper, as presented, seems to require little change, but I suggest that the arguments can be strengthened by reference to some earlier contributions that provided a basis for this break-through contribution.

We thank the Reviewer for the positive comments on our manuscript and constructive suggestions.

1. The work built upon a significant amount of earlier work that identified and summarized the phylogenies of many of the foraminiferal lineages analyzed in the current work. For example, (Kennett and Srinivasan, 1983) graphically illustrated phylogenetic lineages of planktonic foraminifera "Neogene Planktonic Foraminifera- a phylogenetic Atlas"; Hutchinson Ross Pub. Co. These include numerous deeper water lineages discussed in this paper. Also, the characteristic morphology of the taxa are shown in SEM images- a useful resource that helps with the understanding of the evolutionary relationships. Otherwise, most readers will not be familiar with planktonic foraminiferal morphology and the names alone will not click so much from an evolutionary perspective.

Done

2. A crucial part of the hypothesis presented in this paper is that the overall palaeoceanographic process was ultimately driven by global cooling during the late Miocene and later. The evidence for this is not an analytical part of this paper, but instead is provided by reference to previous investigations. However, mentioned references in support of this cooling is limited to just two contributions: Crichton et al (2020) which is largely a modeling effort, and Cramer et al. (2011) (By the way, this reference is absent in the reference list of this paper). There is compelling evidence for Neogene cooling earlier than 7.5 ma which is acknowledged in the statement as " a possible acceleration of global cooling" but this may well have been a distinct climatic threshold event. Evidence for late Miocene cooling has long been known, presented, and discussed in the literature. For this reason, this contribution would be considerably strengthened by acknowledging this. There are numerous examples, but I am most familiar with the evidence gained from studies of sediment cores obtained as the result of major deep sea drilling expeditions in at least three parts of the world ocean. Specific mention of late Miocene cooling is summarized in the following three references (although there are others that the authors might prefer to mention):

Kennett and Von der Borch (1985). Southwest Pacific Cenozoic Paleoceanography. Initial reports of DSDP Project, Vol 90, p 1493.

Kennett and Barker (1990). Latest Cretaceous to Cenozoic Climate and Oceanographic development in the Weddell Sea, Antarctica; an Ocean Drilling Perspective. Proceeding of the Ocean Drilling Program, Scientific Results. Vol. 113, p 937.

Kennett and Exon (2004). Palaeoceanographic Evolution of the Tasmanian Seaway and its Climatic Implications: in Geophysical Monograph Series 151, American Geophysical union, p 345.

All of these contributions also present evidence that this late Miocene and younger global cooling stimulated ocean planktonic biogenic productivity through stimulated ocean overturning circulation. The authors may prefer to reference other such contributions, but in any case, these would strengthen their arguments.

Done, added multiple references including some of those suggested by the reviewer.

3. The wording on a number of the figures is quite small and difficult to read (e.g. Figs 2,3,4,7). It may be possible to make these larger without spoiling the quality of the figures. This is not crucial but something to consider.

Done, we performed the suggested editing on the figures.

4. Minor points:

Line 25- during the latest Miocene… - done

Line 60 - In contrast to planktonic foraminifera… - done

373 - …. impacts the relative abundance between… - done

421-422- clarify and simplify this important sentence - done

427-428- The sentence is clumsy and needs improving - done

430-433- sentence is too complicated and needs rewriting - done

437-438- Simplify the early part of the sentence - done

444- make new sentence, i.e. break up this long sentence. - done

456- what are "foraminiferal prays"? – clarified

5. Check the reference list for completeness.

Done

6. Consider a new title for this paper, since the current one is too general and really does not reflect the importance and focus of this contribution. Example: "The Late Neogene Development of Deep-dwelling Planktonic Ecogroups".

We now propose a new title.

---

## Author Comment (AC2)

Reviewer #2

The manuscript by Boscolo-Galazzo and coauthors is a fantastic contribution to the primary literature investigating the drivers of planktic foraminifera and calcareous nannoplankton through the Neogene. The use of multiple proxies integrated together (models, stable isotopes, biogeography) provides an in-depth look at evolutionary processes and patterns. The paper is very well-structured, well-written, and the figures are robust, informative, and well-designed. The supplementary materials are also very helpful and nicely constructed. I thoroughly enjoyed reading and reviewing this paper, and look forward to citing it in the future!

There are places in the manuscript that could use additional citations and/or references back to the figures that will strengthen the author's discussions and main points. I have also included some additional suggestions regarding including alternative hypotheses for evolutionary types.

We thank the Reviewer for the enthusiastic feedback on our manuscript and her/his constructive and thorough reviews. We provide below point-by-point answers to the reviewer's comments.

Lines 18–20: The statement 'deep-living planktonic foraminiferal species were virtually absent globally during the peak of the middle Miocene warmth' is only true for your tropical sites, correct?

The pie charts (Fig. 5) for Site 516 (mid 1) and Site 1138 (high latitude) do include a higher percentage of twilight planktic foram species compared to your tropical sites, especially Site 1138. This is really cool, and should be highlighted in the text.

Yes and no. It is true that some of the deep-dwelling planktonic foraminifera species occur at Site 516 and 1138 in the middle Miocene, but the depth habitat reconstructions show that at Site 516 they were living above 200 m depth (they plot as undifferentiated grey dots in Fig. 1). For Site 1138 we don't have depth reconstructions available, so unfortunately we cannot tell at what depth they were living at high latitudes. Infact, it is based on their global biogeography and depth reconstructions at Site 516 that we propose the "depth sympatry" mode of speciation for the deep-dwelling taxa from *Hirsutella scitula* starting from the late Miocene (Lines 487-533)

Line 40: Suggest changing 'best studied' to something like 'most thoroughly studied'

Done

Line 55: Change 'reconstruct phylogeny' to 'phylogenetic relationships'

Done

Line 103: First time DSDP, ODP, IODP are mentioned, spell them out

Done

I suggest making Figure 12 into Figure 1, and plotting the locations of your sites onto this map.

We prefer leaving this figure at the bottom of the paper, next to where modelling output are discussed. Also, the locations of the study sites are repeatedly shown in the paleo-maps of Figures 7-11, we think one more map showing them would be redundant.

Section 2.1: Include a sentence here, similar to the sentence in Section 2.2 lines 142–145, where you state what taxonomic references you used. Some refs could include Kennett and Srinivasan, 1983 and the major update to mid-latitude Neogene species Lam and Leckie, 2020a (Micropaleontology), using phylogenetic genus names from Aze et al. (2011).

We thank the Reviewer for noticing we forgot this important peace of information. Added now!

Line 157: Please also cite Matsui et al. (2016), who did an extensive study of changes in depth habit of D. venezuelana through time.

Ok, added.

Matsui, H., Nishi, H., Takashima, R., Kuroyanagi, A., Ikehara, M., Takayanagi, H., & Iryu, Y. (2016). Changes in the depth habitat of the Oligocene planktic foraminifera (Dentoglobigerina venezuelana) induced by thermocline deepening in the eastern equatorial Pacific. Paleoceanography, 31(6), 715-731.

Line 246: Remove 's' from end of 'Figs'

Ok, done.

Lines 312–314: To make this finding more apparent and best connect with biogeography, you could include the date at which the species first appeared before the site, such as (XX Ma, Site 872). In this way, readers can also see the timing of first occurrences across your sites.

Ok, done.

Line 312: In addition to citing Lazarus et al. (1995), the findings of Lam and Leckie (2020b, PLoS ONE) and Jenkins and Srinivasan (1986) also highlight the first appearance of T. truncatulinoides first in the SW Pacific and later in the N Pacific.

Ok, done.

Lines 346–347: It may be worthwhile to include a sentence at the beginning of this section that globoconellids are mid-latitude temperate water dwellers, which will set the stage for this sentence. Could also include citations for such occurrences (e.g., Brombacher et al., 2021).

We said this already in lines 618-619 in the discussion paragraph dedicated to globoconellids, we added the suggested reference therein.

Brombacher, A., Wilson, P. A., Bailey, I., & Ezard, T. H. (2021). The Dynamics of Diachronous Extinction Associated with Climatic Deterioration near the Neogene/Quaternary Boundary. Paleoceanography and Paleoclimatology, e2020PA004205.

Section 4.1: Somewhere in here, should cite Fenton et al. (2016), in which they model environmental controls on diversity through the Cenozoic.

We do not think this paper is relevant here to quote, as they model climate cooling from the early to late Eocene (55-34 Ma), while our work focuses on a much more recent interval (15 Ma to Recent).

Fenton, I. S., Pearson, P. N., Dunkley Jones, T., Farnsworth, A., Lunt, D. J., Markwick, P., & Purvis, A. (2016). The impact of Cenozoic cooling on assemblage diversity in planktonic foraminifera. Philosophical Transactions of the Royal Society B: Biological Sciences, 371(1691), 20150224.

Line 358: Lam and Leckie (2020a, Micropaleo) synonymized G. conoidea with Globoconella miotumida. Throughout the text and on figures, simply change conoidea to miotumida.

Ok, done.

Lines 407–410: Could you make the connection between O2 minimum zones and tight clustering of d18O values more clear here? Or is this sentence about d18O values making another point?

Done, please see lines 415-417.

Lines 451–461: Is this discussion, in part, based on findings of Boscolo-Galazzo, Crichton et al. (2021), or other literature? This part of the discussion would benefit from references to other primary literature and figures in the paper.

Here we are discussing our planktonic foraminifera, nannofossil and modelling data to provide an unifying interpretation for the observed patterns. We added references to figures in the manuscript to make it clearer.

Line 469–470: In addition to Aze et al. (2011), also include Scott et al. (1990) and Kennett and Srinivasan (1983)

Ok, done.

Scott, G. H., Bishop, S., and Burt, B. J., 1990. Guide to some Neogene Globorotalids (Foraminiferida) from New Zealand. New Zealand Geological Survey Paleontological Bulletin 61, Lower Hutt, New Zealand, 176 p.

Kennett, J. P., and Srinivasan, M. S., 1983. Neogene Planktonic Foraminifera: A Phylogenetic Atlas. Hutchinson Ross Publishing Company, Stroudsburg, Pennsylvania, 265 pp.

Lines 554–556: There is literature that points to sustained genetic connectivity among foraminiferal populations in the high latitudes of each hemisphere (bipolarity; e.g., Darling et al., 2000; Norris and de Vargas, 2000), indicating water masses may not be an effective barrier to dispersal. Although allopatric speciation is not impossible in the planktic foraminifera, their high dispersal potential likely creates a scenario where other modes of speciation are more likely (e.g., Norris, 2000). Thus, it may be best to present other alternative modes of speciation that caused the evolution of T. crassaformis from H. cibaoensis, and/or acknowledge the body of literature that supports other modes of speciation in the pelagic realm.

Ok, both done.

Lines 603–604: In addition to Norris et al. (1994) and Aze et al. (2011), please also include the citation Scott et al. (1990).

Ok, done.

Lines 613–635: Also check out evolutionary studies of globoconellids by Norris et al. (1994), Wei and Kennett (1988), and Schneider and Kennett (1996). These papers may be beneficial to this part of your discussion and to integrate with your data.

Norris, R. D., Corfield, R. M., & Cartlidge, J. E. (1994). Evolutionary ecology of Globorotalia (Globoconella)(planktic foraminifera). Marine Micropaleontology, 23(2), 121-145.

Schneider, C. E., & Kennett, J. P. (1996). Isotopic evidence for interspecies habitat differences during evolution of the Neogene planktonic foraminiferal clade Globoconella. Paleobiology, 22(2), 282-303.

We added Schneider and Kennett (1996) as the other references mentioned had already been quoted in the paragraph.

Line 622: Globoconella sphericomiozea was synonymized with G. puncticulata by Lam and Leckie (2020a, Micropaleo), as these species' morphology is not distinguishable.

We actually do not fully agree with this reassessment of the globoconellid taxonomy by Lam and Leckie 2020. In our opinion, specimens identified as *G. inflata* by these authors look more like *G. puncticulata*, based on the number of chambers on the umbilical side, a more evolute umbilical side, and less extensive cortex. Similarly, specimens identified as *G. puncticulata* (e.g. specimens 1-20 Plate 17) look like *G. sphericomiozea*, based on the straight umbilical sutures, open umbilicus, number of chambers on the umbilical side, less rounded, more subacute profiles, as well as the typical pustulous wall texture which characterize Miocene globoconellids as opposed to Pliocene ones. Nonetheless, we acknowledge the this group has a particularly high morphological overlap and intergradation, often making distinction between morphospecies hard.

Lines 640–644, 654, 21: Acknowledge here other potential forms of evolution besides allopatry.

Here, we discuss both allopatry and depth sympatry as possible evolutionary mechanisms.

Figures 7–11: These figures are great, but the map is a bit hard to see. Is it possible to make the maps larger? Likewise, the species names are also small. If the colors on all the pie charts correspond to the same species, could there be just one key per figure showing which pie slice color corresponds to which species?

We now scaled the figures size so that it will be possible to make them bigger on the paper page without losing resolution. Also, the resolution of the figures that will be resubmitted will be remarkably higher also improving the readability of the figures.

Another figure that would aid greatly in visualization, especially for the discussions surrounding biogeography, is one (or a few) showing the ranges of your key species at each site you analyzed. Or, you could have a figure for each species showing its range at each site (arranged from highest to lowest latitude), so readers can compare and contrast first and last occurrence dates at each site and across latitude. Such a figure could go into the Supplemental materials.

We do not have high resolution biostratigraphic data for the study sites. This is because we analysed only one sample per site per time slice, so it is not possible to reconstruct species stratigraphic ranges with the current dataset.